# Optimizing Technical Training for Wheelchair-User Billiard Players through Modified Equipment Implementation

**DOI:** 10.3390/sports12090246

**Published:** 2024-09-06

**Authors:** Viktoriia Nagorna, Artur Mytko, Olha Borysova, Liubov Zhyhailova, Silvio R. Lorenzetti

**Affiliations:** 1Swiss Federal Institute of Sport Magglingen, 2532 Magglingen, Switzerland; artur.mytko@baspo.admin.ch; 2National University of Ukraine on Physical Education and Sport, 03150 Kyiv, Ukraine; borisova-nupesu@ukr.net (O.B.); busya613@gmail.com (L.Z.); 3School of Engineering, ZHAW Zurich University of Applied Sciences, 8401 Winterthur, Switzerland; 4D-HEST, ETH Zurich, 8092 Zurich, Switzerland

**Keywords:** adaptive sport, billiards, wheelchair division, wheelchair users, musculoskeletal disorders, preparation process, para-billiards

## Abstract

This study aims to enhance the effectiveness of the preparation process and the performance of wheelchair users in international billiard competitions through modified equipment. The research methods include analysis and synthesis of the scientific and methodological literature, sociological research methods (questionnaires), expert assessment methods, pedagogical research methods (observation, testing, experimentation), and methods of mathematical statistics. The results of our study are significant: Implementing our developed training program for billiards players with musculoskeletal disorders, utilizing the modified equipment (special mechanical bridge and straps for cue fixation during shots) we created in a pedagogical experiment, demonstrated a probable improvement of 36% in the technical and tactical preparedness of the athletes compared to previous years. This led to a 33% increase in players from the Ukrainian team’s competition performance at the national and European Pool Championships (wheelchair division). In conclusion, implementing our developed training program, accompanied by specialized auxiliary equipment, demonstrated promising results in a pedagogical experiment. These findings underscore the potential of the modified equipment and tailored training programs to optimize sports training for individuals with musculoskeletal impairments in adaptive billiards, contributing to the continued humanization of the sport and offering an effective preparation process for the athletes.

## 1. Introduction

Given the current state of our society, it is imperative to prioritize the needs of individuals facing more significant challenges, such as persons with musculoskeletal disabilities. The variety and versatility of physical exercises in sports activities enable a wide range of positive effects on the psychosocial and physical well-being of individuals with different kinds of disabilities [1]. The scientific literature highlights the prevalence of musculoskeletal impairments, given that they are among the most common conditions. For instance, in the Summer Paralympic Games program, 83% of the competitions are dedicated to athletes with musculoskeletal impairments [1,2].

The analysis of the contemporary literature and advanced experiences reveals that many countries are exploring new avenues of social rehabilitation for individuals with disabilities through sports [3,4,5,6,7]. Special centers for sports rehabilitation provide them with opportunities to engage in various forms of physical activities under careful medical, biological, pedagogical, and psychological supervision, utilizing the latest scientific research on the impact of rehabilitative sports on their health and functionality [6,7,8,9,10,11]. In Ukraine, this experience is relatively new. One primary goal such centers and camps achieve is to involve individuals with musculoskeletal impairments in physical education and sports activities. This indicates the relevance of involving individuals with musculoskeletal impairments in billiards and creating optimal conditions for educational, training, and competitive preparation processes through the modified equipment, thereby enhancing the competitiveness of Ukrainian athletes on the international stage.

An analysis of the current state of the problem highlights the necessity of intensifying the preparation process for these pool players for major international competitions through individualization and optimization of training programs. Additionally, developing the modified equipment that enables the use of the players’ entire technical-tactical arsenal is crucial, as billiards require the precise differentiation of striking force and movement accuracy for the successful execution of its 60 different strokes [12,13]. All official pool tournaments in the wheelchair division have rules about participation eligibility. The nosology explanation is as follows: “The criteria for a player to be eligible for wheelchair competition is that he must be wheelchair mobility dependent for a minimum of 80% of the time. In some cases, a doctor’s letter may be required to determine eligibility”.

Billiards players with musculoskeletal impairments as wheelchair users often face objective challenges in controlling the force and accuracy of their strokes. Accuracy in billiards depends on several parameters: the differentiation of movement amplitude, the force applied to the strike, and the determination of the distance to the target aiming point. Therefore, developing specific equipment to compensate for the inability to execute specific technical maneuvers is a reasonable measure. Additionally, precise alignment of training and competitive loads with the individual capabilities of the athlete is necessary.

This study aims to enhance the effectiveness of the preparation process and the performance of wheelchair users in international billiard competitions through modified equipment.

## 2. Materials and Methods

### 2.1. Study Design

To address the primary objective of our study, we employed a structured problem-solving methodology. Initially, we conducted a bibliometric and scientometric analysis to assess the theoretical landscape of the issue. Subsequently, we compared the most pressing challenges in adaptive billiards as identified through expert evaluations and sociological surveys. The lack of training programs and billiard equipment tailored for wheelchair users prompted us to biomechanically substantiate various stroke techniques and develop specialized training content and auxiliary equipment customized based on the specific musculoskeletal disorders of the athletes. Finally, a pedagogical experiment was conducted to evaluate the effectiveness of these proposed developments. To optimize the writing style of our article, we used the version of ChatGPT based on OpenAI’s GPT-4 architecture. Also, Grammarly (v1.2.95.1470) provided bundled spelling, punctuation, and grammar suggestions.

### 2.2. Participants

The Institutional Ethics Committee approved the research. It was carried out in compliance with the international principles of the Helsinki Declaration of the World Medical Association and by the Law of Ukraine, “Fundamentals of Ukrainian Legislation on Healthcare,” on ethical norms and rules for conducting medical research involving humans. The methodology employed in this study was similar to previous validation studies [12,13,14]. All participants gave written signed consent before any data acquisition.

Our study involved diverse participant groups and methodologies. Surveys gathered insights on adaptive billiards in 37 European countries, with direct communication, including oral interviews and written questionnaires from representatives, coaches, and officials of the European Pocket Billiard Federation and nations participating in the European Pool Championship (2017–2022).

Our pedagogical observation, a unique and crucial part of our research, analyzed the technical actions of 23 top-ranked European pool players with musculoskeletal disorders (38.5 ± 1.87 years men). We focused on the model characteristics of billiard stroke techniques of wheelchair users. This observation led to a significant finding in identifying two distinct types of billiard stroke techniques for wheelchair users. This discovery was so important that in 2023, a biomechanical analysis of these different techniques was carried out as part of the ‘Digital Twin’ project.

Our pedagogical testing, a practical application of our research, assessed the technical-tactical and special physical preparedness of 16 Ukrainian wheelchair-user billiard players (12 male players: 37 ± 5.5 years and four female players: 29.5 ± 2.75 years). This hands-on approach provided valuable data and offered potential strategies for improving these players’ performance in real-game situations. In the medical context, ten athletes from the national wheelchair pool team with musculoskeletal disorders diagnosed with spinal cord injuries at the C5-C6 level were observed. Modified equipment (a unique mechanical bridge and straps for cue fixation during shots) was used during this study, addressing challenges such as complete finger immobility and an inability to perform specific movements.

### 2.3. The Bibliometric and Scientometric Analysis

Bibliometric methodologies were employed to systematically retrieve highly cited papers in adaptive sports sciences, encompassing 2003 to 2024. Multiple databases, including Scopus, the Clarivate Analytics Web of Science Core Collection, Google Academy, Webometrics, and Perplexity (https://www.perplexity.ai/search/, accessed on 26 April 2024), were scrutinized to ensure a comprehensive search strategy. The keywords employed for information retrieval were strategically chosen to capture the breadth of the research domain, including adaptive sports, adaptive billiards, musculoskeletal disorders, wheelchair users, technical-tactical preparation in billiards, modified equipment, special auxiliary equipment, sports training optimization, and performance improvement. Theoretical analysis of the scientific-methodological literature and documentary materials allowed for a detailed assessment of the state of the scientific problem, justification of the relevance of the research topic, formulation of objectives, and selection of appropriate research methods for billiard players with musculoskeletal impairments.

### 2.4. The Methods Used Are Expert Evaluations and Sociological Surveys

The Sociological Survey in our study served as a comprehensive method for gathering information on the state of adaptive billiards development in European countries. This involved direct and indirect communication channels, including oral interviews and written questionnaires. The participants in this survey comprised representatives from each member country of the European Pocket Billiard Federation (EPBF), as well as coaches and officials from nations actively involved in the European Pool Championship from 2017 to 2022. The survey reached 37 national federations affiliated with the EPBF, with responses obtained from 20 of them providing extensive insights into the landscape of adaptive billiards within their respective countries. The findings revealed a notable interest among individuals with various disabilities in billiards. However, challenges were identified, indicating that federations or billiard clubs often need help to offer favorable conditions for training and competitive participation for this demographic.

A comprehensive list of primary challenges impeding the development of adaptive billiards was extrapolated from respondents’ answers in 2023 to analyze the survey data systematically. These challenges were subsequently categorized into six main reasons contributing to the observed low activity levels among individuals with disabilities in adaptive billiards. Subsequently, these identified factors were incorporated into a matrix of experts’ assessments involving 20 experts. This matrix facilitated the determination of the most critical elements negatively influencing the integration of people with musculoskeletal disorders into the sports realm, providing valuable insights for further investigation and intervention strategies. A meticulous analysis of expert assessments was conducted to ascertain the collective expert opinion on prevalent issues in adaptive billiards (refer to Table 1) that impede the effective progression of pool training and competitive performance for wheelchair users. To facilitate this analysis, a matrix delineating each expert’s ratings of ranked factors was devised (refer to Table 2).

Within this matrix, experts identified and ranked the tools they deemed most significant for adverse manifestations in adaptive billiards. The ranking was structured in descending order of influence, with experts assigning positions from 1 to 6 based on their perceived impact.

### 2.5. Pedagogical Observation and Testing

We conducted pedagogical observation, which included analyzing the technical actions of highly skilled billiard players from Europe and the top-ranked Ukrainian athletes with musculoskeletal impairments, involving 23 athletes, to obtain information on the model characteristics of the main stroke techniques for wheelchair pool players.

The pedagogical observation method identified two primary techniques wheelchair-user players employ in executing a basic billiard shot. Following this, a biomechanical analysis of these two techniques was conducted using the innovative “OpenCap” development [15]. For biomechanical analyses using OpenCap (opensimModel: LaiArnoldModified2017_poly_withArms_weldHand, posemodel: openpose, augmentermodel: v0.2), the system utilized paired cameras from two iOS devices connected to a web application operating on a standard laptop, which recorded videos at a rate of 60 Hz.

Pedagogical testing was conducted to determine the technical-tactical and specific physical preparedness levels of 16 billiard players with various musculoskeletal impairments. The tests encompassed a series of technical and tactical exercises (Figure 1):

Basic shots without cue ball positioning for the next ball—“cue-ball on the spot” for each shot (a)—maximum 15 points for one exercise, one pocketed object ball gives 3 points; difficult shots, like bank shots, without cue ball positioning for the next ball—“cue-ball in hand” for each shot (b)—maximum 15 points for one exercise, one pocketed object ball gives 3 points; basic shots with cue ball positioning in a specific area (c). The exercise involves executing basic shots with cue ball positioning in three specific table areas, consisting of five follow shots, five stop shots, and five draw shots. Each exercise has a maximum score of 15 points. One point is awarded for each pocketed object ball made with follow, stop, or draw shots, depending on the aim of the cue ball position; performing difficult shots involving cue ball positioning to reach the next object ball, such as in the “Replacement Line” exercise (d), where the cue ball is initially placed “in hand” for the first shot but subsequently played from its stopping position after pocketing an object ball. The objective is to pocket all object balls without touching others until a miss occurs, with a maximum score of 15 points, where each ball pocketed earns 3 points.

Following the logic of defining a similar indicator in other sports, a billiard player’s special physical preparedness encompassed values corresponding to the combination of technical and tactical preparedness and individual game performance.

In the context of the medical background of musculoskeletal disorders observed in a group of 10 athletes from the national wheelchair pool team who participated in our experimental study (using a modified “mechanical bridge” and special straps for cue fixation during the stroke), a diagnosis of spinal cord injury at the C5-C6 level was established. The individuals exhibited complete immobility of the fingers and functional extensor activity in the wrist but could not perform flexion movements.

### 2.6. Biomechanical Analysis

Biomechanical analysis of two distinct billiard stroke techniques for wheelchair-user players was conducted using contemporary methodologies, such as OpenCap (Figure 2). For OpenCap (opensimModel: LaiArnoldModified2017_poly_withArms_weldHand, posemodel: openpose, augmentermodel: v0.2), the setup involved synchronizing cameras from two iOS devices with a web application running on a standard laptop, capturing videos at a rate of 60 Hz. The OpenCap methodology involved several steps for estimating movement dynamics from videos, including camera calibration, video collection and processing, marker position estimation, kinematics estimation, and the generation of physics-based dynamic simulations of movements. This comprehensive pipeline was implemented in Python (v3.7.10), and the OpenCap web application provided a user-friendly interface, guiding users through each step. Cloud instances were utilized for the computational processes.

### 2.7. Statistical Analysis

The level of specialized physical preparedness was calculated using the following formula:(1)$$SPhP=\frac{\sum B_{player}^{1-4}\cdot 100}{\sum B_{max}^{1-4}}$$
where ∑$B_{player}^{1-4}\;$ is the sum of points for 4 test exercises performed by wheelchair user pool players, and ∑$B_{max}^{1-4}\;$ is the maximum possible sum of points for 4 test exercises to determine the level of specialized physical preparedness.

The formula for calculating the effectiveness of competitive performance was as follows:(2)$$R_{new}=R1\geq R2\Rightarrow R_{1}+M\cdot V\cdot 25{\left(\frac{R1}{R2}\right)}^{-1.7}R1\leq R2\;\Rightarrow R_{1}+M\cdot V\cdot \left(25-9ln\left(\frac{R1}{R2}\right)\right)$$
(3)$$\mathrm{where}\;M=\frac{\sum R^{1-16}tournament}{\sum R^{1-16}rating};\ldots V=\frac{S_{1}-S_{2}}{\sqrt{5\cdot max\left(S_{1};S_{2}\right)}}$$

R_1_ and R_2_ are the players’ ratings;

M is the tournament coefficient (the ratio of the sum of points of the top 16 players in the tournament to the sum of points of the top 16 players in the rating); the minimum value is 0.5;

V is the significance of victory;

S_1_, S_2_ are the number of wins by the 1st and 2nd players in the match;

max (S_1_, S_2_) is the match format (the number of wins required); the larger the format, the more points are at stake.

Mytko modified the formula for billiards [14].

During the statistical processing of the research, the effectiveness of the competitive activity was evaluated according to the indicator of the probability of events. According to the probability theory, we determined the possibilities of adequate performance of specific types of shots in billiards when using innovative aids developed for people with injuries of the musculoskeletal system [16].

The degree of agreement between experts’ answers was determined using Kendall’s concordance coefficient (W). Kendall’s concordance coefficient was calculated using the following formula:(4)$$W=\frac{12\cdot S}{m^{2}\left(n^{3}-n\right)}$$
where S is the sum of the squares of the deviation of the estimate from the mean value:(5)$$S=\sum_{i=1}^{n}{\left(\left(\sum_{j=1}^{m}x_{ij}\right)-\bar{x}\right)}^{2}$$
where m is the number of experts;

n is the number of examination objects,$x_{ij}$—i-th evaluation of the j-th expert;

$\bar{x}$ is the average score given by m experts for all n objects of examination, which is determined by the following formula:(6)$$\bar{x}=\frac{m\cdot \left(n+1\right)}{2}$$
where m is the number of experts; n is the number of examination objects.

The coefficient of significance of each factor established as a whole by the group of experts is determined by the following formula:(7)$$K_{j}=\frac{m\cdot n-x_{j}}{0.5\cdot m\cdot n\cdot \left(n-1\right)}$$

Determination of the normative coefficient of significance ($K_{N}$), which is the reciprocal of the number of ranked factors:(8)$$K_{N}=\frac{1}{n}$$

Respondents’ answers to the questionnaire questions were determined by points, which assessed the probability of the obtained results using non-parametric statistical methods. Then, we used the Mann–Whitney U-test, the statistical significance of which was checked using the χ^2^ test.

The significant finding of difference from constants in identifying two distinct types of billiard stroke techniques of 23 wheelchair users (one sample case): Type of power analysis—Sensitivity; Effect size (Cohen’s d)—0.5; α err prob—0.05; Power (1-β err prob)—0.80.

The significant finding of difference from constants in identifying the technical-tactical and special physical preparedness level of 16 wheelchair users before and after modification equipment (one sample case): Type of power analysis—Sensitivity; Effect size (Cohen’s d)—0.5; α err prob—0.05; Power (1-β err prob)—0.65.

Finally, all statistical hypotheses were tested at the α = 0.05 significance level (*p* < 0.05), and mathematical and statistical processing and data analysis were conducted using Statistica (Statsoft, version 7.0) and Microsoft Excel 2010.

## 3. Results

The literature review [17,18,19,20,21] encompasses diverse studies on physical activity and mobility in various populations, emphasizing individuals with spinal cord injuries and manual wheelchair users. These studies cover topics such as the humeral elevation workspace during daily life in adults with spinal cord injury who utilize manual wheelchairs, comparing them to age and sex-matched able-bodied controls. Additionally, the review explores using a neural network model with inertial body-worn sensors to estimate manual wheelchair-based activities in free-living environments. The broader perspective of public health and physical activity is also considered, highlighting the role of legislative and policy initiatives in increasing physical activity levels, with potential implications for individuals with disabilities. Lastly, the discussion extends to the concept of inclusion in sports, particularly addressing disability and tournament participation. This diverse set of studies collectively contributes to understanding physical activity, mobility, and inclusive practices in sports and exercise contexts. The insights provided are valuable for wheelchair users and their coaches. Combined with expert surveys (conducted among 37 representatives from European countries, with 20 specialists selected for expert evaluation), they confirm the need for objectively assessing athletes’ conditions during training and competitive processes. This enables adjustments to training tasks for individualization, emphasizing the importance of special equipment and inventory for wheelchair users (Figure 3).

To record the kinematic parameters of motor actions during the execution of the same type of billiard shot but using two different techniques, the OpenCap application was employed, focusing on joints with one degree of freedom, such as the shoulder and elbow of the player’s dominant hand. The trajectory of the point relative to the external coordinate system was constructed, namely, the kinematic diagram of two motor actions—the player’s stroke arm in the “vertical” and “horizontal” shot techniques. Anatomical marker positions were estimated from the recorded videos to compute joint kinematics using OpenSim’s Inverse Kinematics tool and the scaled musculoskeletal model. The resulting 3D kinematics were visualized in the web application. We determined there was a reasonably close correlation (r = 0.98) between indicators of straight-line motion of the cue during a shot by a billiards player using two different techniques (*p* value = 0.01): with the forearm in the vertical plane and with the forearm in the horizontal plane (Figure 4).

Pedagogical observations and testing of technical preparedness indicators before the implementation of the experiment yielded the following results: basic shots without cue ball positioning for the next shot were executed successfully by 100% of the athletes, while difficult shots without cue ball positioning for the next shot were executed successfully by 85% of the athletes; basic shots with cue ball positioning were executed successfully by 87.5% of the athletes, and difficult shots with cue ball positioning for the next object ball were executed successfully by only 56.25% of the athletes. Such low indicators of the efficiency of the technical preparedness of most players with musculoskeletal disorders compared to non-disabled individuals can be explained by low maneuverability in the choice of possible types of strokes.

The low percentage distribution in the effectiveness of difficult shots, which involve cue ball positioning for the next object ball, performed by athletes is understandable. In a game situation, tactics may necessitate positioning the cue ball at a location on the table where it would be inconvenient for a player who uses a wheelchair to execute a shot.

The level of specialized physical preparedness was calculated using the formula we modified specifically for billiards. We obtained an average coefficient of the specialized physical preparedness level for wheelchair users who are pool players, denoted as SPhP = 28 ± 4.42. Furthermore, the specialized formula for billiards was applied to compute the coefficient of effectiveness of competitive performance at national and international levels, yielding R_new_ = 20 ± 8.21.

Expert surveys and biomechanical analyses of athletes’ movements during technical maneuvers have necessitated the development of special equipment for billiards players with musculoskeletal impairments. The research findings have facilitated the formulation of individualized training plans for these players by implementing the modified equipment.

To increase the range of technical and tactical skills and improve performance in competitive activities, we have developed special additional equipment for billiards players with disabilities:Special straps for cue fixation during shots for athletes with spinal cord injuries in the neck region.A modified “mechanical bridge” to enable the execution of a maximum number of technical shots.

The developed special equipment was implemented into the training process of the Ukrainian national team in billiards for athletes with musculoskeletal injuries. Pedagogical monitoring was carried out from 2018 to 2023, during which the following indicators were taken into account: the quality of performing test exercises on technique (basic and advanced), technical-tactical and special physical preparedness (differentiation of muscle efforts, coordination of movements, special endurance), and the results in major competitions.

The practical implementation of individualized training programs for billiard players, complemented by the concurrent utilization of modifications such as the adapted “mechanical bridge” (refer to Figure 5 and Figure 6) and specialized straps for cue fixation during strokes for athletes with musculoskeletal injuries in the cervical spine (refer to Figure 7 and Figure 8), has facilitated notable improvements in both technical and tactical training. Observable changes were noted following the second round of testing, resulting in an average coefficient of specialized physical preparedness level for wheelchair users, SPhP = 72 ± 3.84, with significant improvement (*p* < 0.05). Additionally, the coefficient of effectiveness of competitive performance at both national and international levels yielded R_new_ = 60 ± 5.12, with significant improvement (*p* < 0.05). This indicates a substantial 36% increase in the volume of specialized physical preparation and a significant 33% growth (*p* < 0.05) in performance outcomes at pool competitions.

Description of the design of the modified “mechanical bridge” for billiard players with musculoskeletal disabilities in the cervical spine:

The modified “mechanical bridge” for billiard players with musculoskeletal disabilities can have a unique design that allows for the stable fixation of the cue during the stroke.

The modified “mechanical bridge” consists of a base and support that provide stability during the game. The base can be flat and sufficiently large to ensure reliable support. The support should be located on one side of the base and have a unique shape for cue fixation. A soft or adjustable holder for cue fixation can be placed on the support. Considering the player’s limited mobility, this holder should provide stable and secure cue retention during the stroke. Various materials such as wood, sturdy plastic, metal, or a combination of different materials can create the modified “mechanical bridge.” It is essential to use materials that ensure stability and durability.

Some modified “mechanical bridges” can have adjustable features to accommodate each user’s individual needs. For example, the support’s height, angle of inclination, or width can be adjusted based on the user’s requirements.

The implementation of authorial innovative developments (unique straps for cue fixation during the stroke for athletes with musculoskeletal injuries in the cervical spine and a modified bridge) and a specialized training program for billiard players with musculoskeletal injuries contributed to the significant improvement (*p* < 0.05) of the following indicators:

Technical preparedness increased by 25%.

Technical-tactical preparedness increased by 30%.

Special physical preparedness increased by 36%.

Performance in competitions improved by 33%.

## 4. Discussion

The findings from our study underscore the critical role of objective assessments and specialized equipment for wheelchair users in sports. They align with the results of a specialized questionnaire commissioned by the Swedish Billiards Federation. This survey revealed significant disparities in the support and opportunities provided by national federations for athletes with disabilities participating in para-billiards activities.

While some national federations demonstrated a solid commitment to inclusivity by offering dedicated activities and competitions for para-athletes, others needed to provide the necessary support and resources. This disparity highlights an urgent need for standardization and enhancement of programs designed to facilitate the participation of athletes with disabilities in billiards.

The challenges identified in training athletes with musculoskeletal disorders, such as spinal cord injuries, emphasize the importance of tailored approaches in both assessment and training methodologies. Our study strongly advocates using objective assessments to inform personalized training programs, addressing individual athletes’ unique needs and limitations. Such targeted training is crucial for maximizing the potential of athletes with disabilities, ensuring they receive the most effective support possible.

Furthermore, the development and use of technical utilities and appliances represent promising avenues for improving accessibility and performance in para-billiards. However, as noted by some national federations, more awareness about these tools is needed, underscoring the need for greater investment in and dissemination of innovative solutions. Raising awareness and investing in such technologies are essential to enhancing inclusivity and competitiveness in the sport.

Our study and the specialized questionnaire findings highlight the necessity of comprehensive strategies to promote the participation and success of athletes with disabilities in billiards. Addressing the identified challenges and leveraging advancements in assessment tools and specialized equipment can make the sport more inclusive and equitable for all participants, regardless of their physical abilities.

Our research has established the scientific and practical relevance of the training program and accompanying tools we developed, which aim to enhance the effectiveness of competitive activities in the national adaptive billiards (pool) team—currently without analogs. These developments lay a solid foundation for implementing auxiliary technical means in adaptive sports, which can be produced using 3D medical use or 3D printers for prosthetics [22,23,24,25,26,27,28,29].

In the context of innovative developments for wheelchair-user billiard players, a modified “mechanical bridge” manufacturing process can be optimized using 3D printing technology. This modified “mechanical bridge” can be customized to have a unique shape and size, allowing players with injuries to the musculoskeletal system in the neck region to stabilize the cue better during a shot. Individually tailored “bridges” help ensure accuracy and stability in shots, thereby improving technique and game tactics.

Modifying equipment is essential in adaptive sports, enabling athletes with disabilities to participate fully and compete at high levels. Tailoring gear to meet individual needs opens the door for more people to engage in physical activities and competitive sports who might otherwise be excluded. This increased accessibility promotes inclusivity and equal opportunities within the sporting world [30,31,32,33].

Modified equipment maximizes an athlete’s abilities and compensates for physical limitations. Appropriately modified equipment ensures the safety and comfort of athletes with disabilities, helping to prevent injuries by accommodating individual physical needs. Comfortable, well-fitted gear allows athletes to focus on their performance rather than struggling with unsuitable equipment.

Moreover, the right adaptive equipment can significantly boost an athlete’s self-confidence. It provides physical support and nurtures a belief in one’s abilities. This psychological empowerment is crucial for athletes to set and achieve ambitious goals, regardless of their physical challenges.

The field of adaptive sports equipment is driving technological innovation. Ongoing research and development in materials science, biomechanics, and digital technology are pushing the boundaries of what is possible in prosthetics and other adaptive gear. This innovation benefits athletes and has broader applications in medical and assistive technologies [34].

The equipment modification is fundamental to the success and growth of adaptive sports. It opens doors for participation, enhances performance, ensures safety, boosts confidence, drives innovation, and promotes a more inclusive sporting environment for all athletes.

## 5. Conclusions

The literature review provides a comprehensive analysis of studies focusing on physical activity and mobility, particularly among individuals with spinal cord injury and manual wheelchair users. These insights enhance our understanding of inclusive practices in sports and exercise. Expert surveys emphasize the need for objective assessments of athletes’ conditions, facilitating tailored training approaches and highlighting the importance of specialized equipment for wheelchair users.

Employing the OpenCap application, we examined kinematic parameters during billiard strokes, revealing a close correlation (r = 0.98) of cue movement between the two techniques. However, due to tactical constraints, only 56.25% of athletes successfully executed shots from uncomfortable positions. The biomechanical analysis led to developing specialized equipment for players with musculoskeletal impairments, enhancing training plans. Implementing individualized training programs, alongside modifications such as the “mechanical bridge” and specialized straps, resulted in a 36% increase in specialized physical preparation and a significant 33% growth in performance outcomes at international competitions.

The scientific and applied relevance of the innovations developed to enhance the effectiveness of the national team’s training and competitive activities in adaptive billiards (pool) has been proven. These developments can be applied to other forms of adaptive sports worldwide.

## Figures and Tables

**Figure 1 sports-12-00246-f001:**
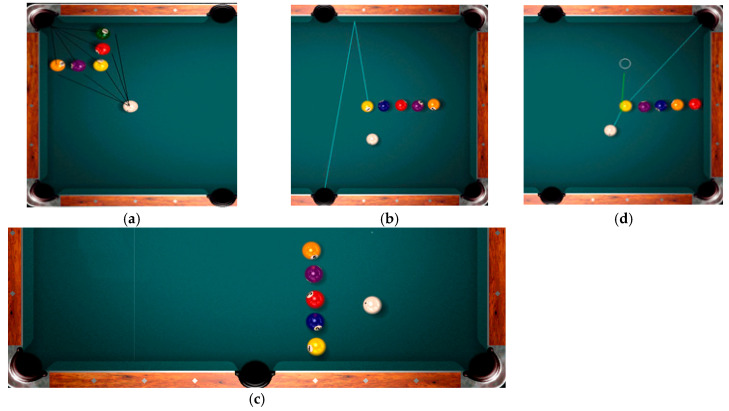
The pedagogical tests for billiard players with various types of musculoskeletal impairments (n = 16) consist of technical and tactical exercises: basic shots without cue ball positioning for the next ball (**a**), difficult shots without cue ball positioning for the next ball (**b**), basic shots with cue ball positioning in a specific area (**c**), difficult shots involving cue ball positioning to the next object ball (**d**).

**Figure 2 sports-12-00246-f002:**
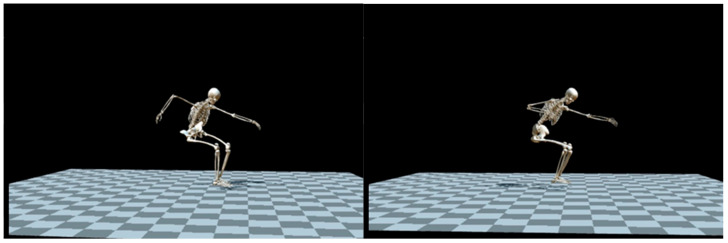
Biomechanical analysis of wheelchair-user players’ billiard stroke techniques using the innovative “OpenCap” development.

**Figure 3 sports-12-00246-f003:**
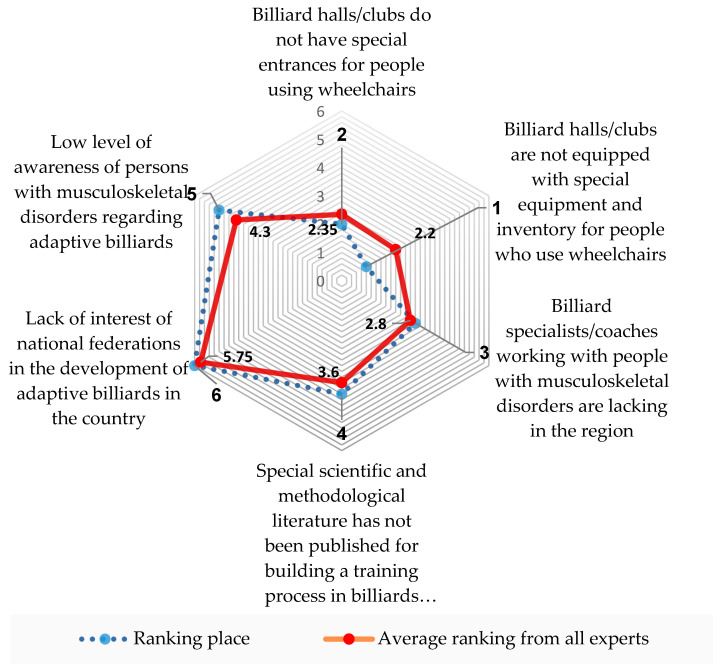
Analysis of expert (m = 20) assessments on prevalent issues in adaptive billiards impeding the effective progression of pool training and competitive performance for wheelchair users.

**Figure 4 sports-12-00246-f004:**
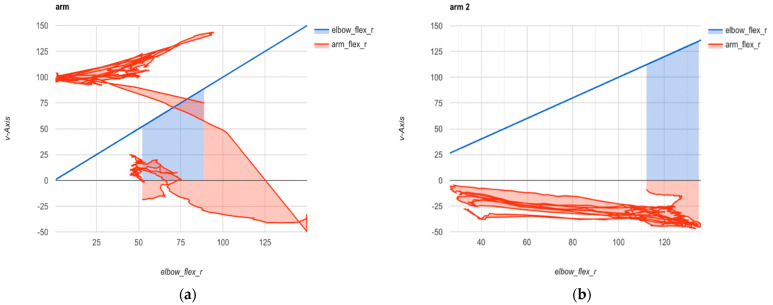
The kinematic diagram of two motor actions—the player’s stroke arm in the “vertical” (**a**) and “horizontal” (**b**) shot techniques for wheelchair users.

**Figure 5 sports-12-00246-f005:**
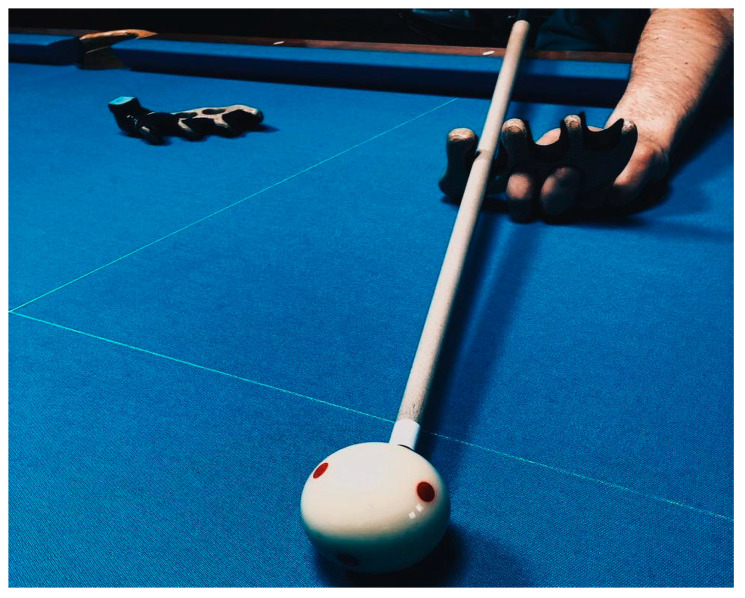
The modified “mechanical bridge” in position for basic shots in billiards during stroke for athletes with musculoskeletal injuries.

**Figure 6 sports-12-00246-f006:**
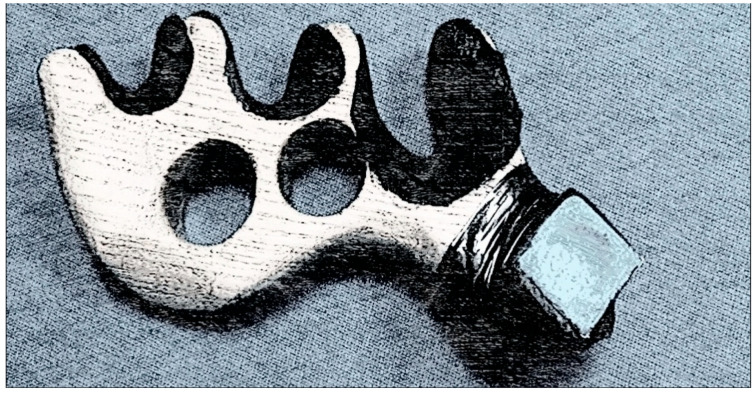
The modified “mechanical bridge” for wheelchair billiard players.

**Figure 7 sports-12-00246-f007:**
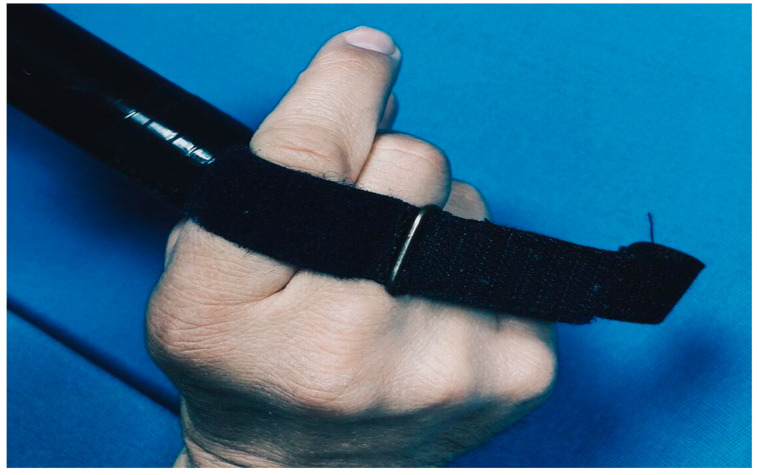
Special belts for wheelchair billiard players to fix the cue during stroke for athletes with musculoskeletal injuries.

**Figure 8 sports-12-00246-f008:**
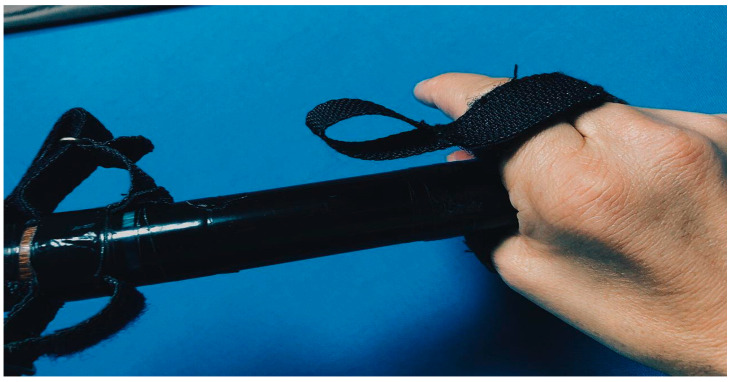
Special belts are used to fix the cue during impact for athletes with injuries to the musculoskeletal system in the neck.

**Table 1 sports-12-00246-t001:** Numbering of experts’ factors.

Name of Factor	Factor Number
Billiard halls/clubs do not have special entrances for people using wheelchairs	1
Billiard halls/clubs are not equipped with special equipment and inventory for people who use wheelchairs	2
Billiard specialists working with people with musculoskeletal disorders must be improved in the region	3
Unique scientific and methodological literature has yet to be published for building a training process in billiards for persons with musculoskeletal disorders	4
Lack of interest of national federations in the development of adaptive billiards in the country	5
Low level of awareness of persons with musculoskeletal disorders regarding adaptive billiards	6

**Table 2 sports-12-00246-t002:** Matrix of experts’ assessments.

Expert Number (m)	Factor Number (n)						$x_{i}$
	1	2	3	4	5	6	
1.	2	1	3	4	6	5	21
2.	4	2	1	6	5	3	21
3.	1	3	4	5	6	2	21
4.	4	1	3	2	6	5	21
5.	1	2	3	4	6	5	21
6.	4	2	1	5	6	3	21
7.	3	2	1	5	6	4	21
8.	2	1	3	4	6	5	21
9.	1	6	5	3	2	4	21
10.	2	3	1	4	6	5	21
11.	1	3	2	4	6	5	21
12.	4	1	3	2	6	5	21
13.	1	2	3	5	6	4	21
14.	4	1	3	2	6	5	21
15.	1	2	4	3	6	5	21
16.	3	1	4	2	6	5	21
17.	4	2	1	3	6	5	21
18.	2	5	4	3	6	1	21
19.	1	3	4	2	6	5	21
20.	2	1	3	4	6	5	21
$x_{j}$	47	44	56	72	115	86	420
$K_{j}$	0.24 *	0.25 *	0.21 *	0.16	0.02	0.11	0.17
W							0.71

*—the factor is significant, as the value of the coefficient significance is equal to or exceeds the value of the normative coefficient ($K_{j}\;$≥ 0.17) because of the normative coefficient of significance $K_{N}$ = 0.17.

## Data Availability

The data supporting this study’s findings are available from the corresponding author, V.N., upon reasonable request. However, the data are not publicly available due to privacy and ethical restrictions.

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
