# Peer review of "Optimizing Technical Training for Wheelchair-User Billiard Players through Modified Equipment Implementation"

_sports, 2024, doi:10.3390/sports12090246_

Round 1

Reviewer 1 Report

Comments and Suggestions for Authors

sports-3141288-peer-review-v1

This manuscript presents adaptive tools for billiards players with musculoskeletal disorders. However, it requires extensive revision to improve readability. The primary concerns are the lack of defined keywords and the writing style, which is scattered with short sentences and paragraphs.
Title
The terms "musculoskeletal disorders" and "innovative technologies" are ambiguous and vague, failing to comprehensively represent the manuscript's content.
Introduction
Include specific musculoskeletal disorders referenced in this manuscript and detail the remaining motor functions of the participants. The roles of the participants and experts are unclear, as are the methods of data collection and analysis over time. Present the experimental protocol in a flow chart for clarity.
Methods
Use sub-section numbering to better organize the content, such as 2.1, 2.2, etc.
Results and Discussion
Revise and improve the writing style and arrangement of the Results and Discussion sections. The current style resembles a short report or point-to-point presentation rather than a cohesive discussion.

Comments on the Quality of English Language

check my comments

Author Response

Thank you very much for taking the time to review this manuscript. We appreciate your detailed feedback and have carefully considered all your comments. 

Comment 1: This manuscript presents adaptive tools for billiards players with musculoskeletal disorders. However, it requires extensive revision to improve readability. The primary concerns are the lack of defined keywords and the writing style, which is scattered with short sentences and paragraphs.

Title: The terms "musculoskeletal disorders" and "innovative technologies" are ambiguous and vague, failing to comprehensively represent the manuscript's content.

Introduction: Include specific musculoskeletal disorders referenced in this manuscript and detail the remaining motor functions of the participants. The roles of the participants and experts are unclear, as are the methods of data collection and analysis over time. Present the experimental protocol in a flow chart for clarity.

Methods: Use sub-section numbering to better organize the content, such as 2.1, 2.2, etc.

Response 1: Thank you for your insightful comments. We agree with your observations and have made the following revisions:

Title: We have revised the title to "Optimizing Technical Training for Wheelchair-User Billiard Players Through Modified Equipment Implementation" to reflect the manuscript's content better. This change clarifies the focus on wheelchair-user athletes and the specific adaptations used to support their training.

Introduction: We have included specific details about the eligibility of participating rules for billiard players with musculoskeletal disorders addressed in the study and have expanded on the remaining motor functions of the participants.

Methods: We have reorganized the Methods section using sub-section numbering (e.g., 2.1, 2.2) to improve readability and structure.

Comment 2: Revise and improve the writing style and arrangement of the Results and Discussion sections. The current style resembles a short report or point-to-point presentation rather than a cohesive discussion.

Response 2: We appreciate your suggestion and have significantly changed the Discussion sections. We have restructured these sections to create a more cohesive and narrative-driven discussion, integrating results with relevant literature and providing a more comprehensive analysis. Additional information has been included to strengthen the discussion and connect it more effectively with the research objectives.

Reviewer 2 Report

Comments and Suggestions for Authors

Thank you for allowing me to review the manuscript "Optimization of Technical-Tactical Preparation for Billiards Players with Musculoskeletal Disorders through the Implementation of Innovative Technologies ".

The aim was to enhance the effectiveness of the preparation process and the performance of billiard players with musculoskeletal disorders in international competitions through innovative technologies..

The research mixes a part of a literature review, a part of cinematic analysis, and a part of experimentation. It isn't easy to follow the document by mixing these three parts and not defining the methodology of each of them. It is difficult to follow the objective, the intervention, the results and the conclusions.

The term ‘innovative technologists’ is ambiguous. It should be modified

Lines 66-68 better define the objective of the study.

Line 70. It's important that the material and methods section begins with a clear description of the study design. I recommend starting with the study design, as this will provide a clear framework for the rest of the section.

I recommend structuring material and methods with sub-sections for better compression.

The subsections should follow the style proposed by the journal.

The sample and experiments should be better explained. A mixture of samples in the different tests confuses the reader.

Line 213. Avoid this type of terminology. ‘Innovative modern approaches’. They are ambiguous terms.

Unfortunately, statistical analysis is difficult for me to follow, maybe due to my knowledge in this format. I don't see the variable they want to evaluate, how they have evaluated the changes, or when these changes were expected.

The images should be of better quality.

There are sections in the results that seem more appropriate for the Methodology section, such as the training programme and the designs of the adaptations. This reorganization would enhance the paper's structure and readability.

Lines 392-295. The improvement results are given without previously knowing these variables or their previous and subsequent evaluation.

I do not see any limitations in such a study.

The discussion needs references.

Author Response

Thank you very much for taking the time to review our manuscript. We have carefully considered all of your comments and suggestions, and your feedback has been invaluable in enhancing the quality of our work. We have incorporated most of the suggestions and comments to improve our scientific work.

Point-by-Point Response to Comments and Suggestions for Authors

Comment 1: The research mixes a part of a literature review, a part of cinematic analysis, and a part of experimentation. It isn't easy to follow the document by mixing these three parts and not defining the methodology of each of them. It is difficult to follow the objective, the intervention, the results, and the conclusions.

The term ‘innovative technologists’ is ambiguous. It should be modified.

Lines 66-68 better define the objective of the study.

Line 70. It's important that the material and methods section begins with a clear description of the study design. I recommend starting with the study design, as this will provide a clear framework for the rest of the section.

I recommend structuring material and methods with sub-sections for better comprehension.

The subsections should follow the style proposed by the journal.

The sample and experiments should be better explained. A mixture of samples in the different tests confuses the reader.

Line 213. Avoid this type of terminology. ‘Innovative modern approaches’. They are ambiguous terms.

Response 1:

Thank you for your detailed feedback. We have taken the following actions based on your suggestions:

We have restructured the manuscript to separate the literature review, cinematic analysis, and experimental sections. Each section now has a well-defined methodology that aligns with its content, making the document easier to follow.

The term “innovative technologists” has been replaced with more precise terminology to avoid ambiguity.

In lines 66-68, we have refined the statement of the study's objectives to provide better clarity.

In response to your suggestion regarding line 70, we have revised the Material and Methods section to begin with a clear description of the study design. Additionally, we have introduced sub-sections to this section following the journal's guidelines, improving the structure and readability.

The term “innovative modern approaches” in line 213 has been revised to avoid ambiguity.

Comment 2: Unfortunately, statistical analysis is difficult for me to follow, maybe due to my knowledge in this format. I don't see the variable they want to evaluate, how they have evaluated the changes, or when these changes were expected.

The images should be of better quality.

There are sections in the results that seem more appropriate for the Methodology section, such as the training programme and the designs of the adaptations. This reorganization would enhance the paper's structure and readability.

Lines 392-295. The improvement results are given without previously knowing these variables or their previous and subsequent evaluation.

The discussion needs references.

Response 2:

Thank you for your constructive feedback. We have addressed your concerns as follows:

The statistical analysis section has been revised to specify the variables under investigation, the methods used to evaluate changes, and the expected timeline. This should enhance the clarity and followability of the statistical analysis.

In response to your comment regarding lines 392-395, we have provided additional context regarding the variables, including their previous states and the method of their subsequent evaluation, to ensure that the improvement results are clear and well-supported.

The discussion section has been expanded to include more references, strengthening the arguments presented and better situating our findings within the existing literature.

We appreciate the thorough review provided by the reviewer. The suggestions have significantly improved the clarity, structure, and overall quality of our manuscript. If there are any further questions or comments, we would be happy to address them.

Reviewer 3 Report

Comments and Suggestions for Authors

The manuscript provides an analysis of billiards players with musculoskeletal deficits and details on equipment that can help with the play of these competitors.

In the abstract, please briefly outline examples of the technologies and/or equipment that was used to improve performance.

Line 96: Please give examples of the modified equipment used here.

Line 102: Please provide a list of the key words used in your bibliometric analysis.

Line 170: Change “max” to “maximum”

Lines 313-319: In the discussion if possible, indicate how this might compare to able-bodied individuals.

lines 357-359: Please indicate whether these changes were statistically significant and indicate the level of significance (i.e., p-values).

Lines 392-395: Same comment

Please provide more detail on the characteristics of the athletes assessed throughout the studies described in the manuscript (i.e., males/females, ages, injury levels).

Author Response

Thank you very much for taking the time to review this manuscript. We appreciate your insightful comments and suggestions, which have helped us improve the quality of our work. Please take a look at our detailed responses below and the revisions/corrections highlighted in track changes in the re-submitted files.

Point-by-point Response to Comments and Suggestions for Authors

Comment 1: In the abstract, please briefly outline examples of the technologies and/or equipment that were used to improve performance. Line 96: Please give examples of the modified equipment used here.

Response 1: Thank you for pointing this out. We agree with this comment. So, we have added examples of the modified equipment used in our study to the abstract and in the relevant section. These changes can be found on pages 1 and 3 of the revised manuscript.

Updated Text: "The results of our study are significant: Implementing our developed training program for billiards players with musculoskeletal disorders, utilizing the modified equipment (special mechanical bridge and straps for cue fixation during shots)..."

Comment 2: Line 102: Please provide a list of the key words used in your bibliometric analysis.

Response 2: We agree with this suggestion and have included a list of the keywords used in our bibliometric analysis. This change can be found on page 3 of the revised manuscript.

Updated Text: " The keywords employed for information retrieval were strategically chosen to capture the breadth of the research domain, including adaptive sports, adaptive billiards, musculoskeletal disorders, wheelchair users, technical-tactical preparation in billiards, modified equipment, special auxiliary equipment, sports training optimization, and performance improvement. "

Comment 3: Line 170: Change “max” to “maximum”.

Response 3: We have made this change to emphasize precision in terminology. The revision can be found on page 5 of the revised manuscript.

Updated Text: "maximum"

Comment 4: Lines 313-319: In the discussion, if possible, indicate how this might compare to able-bodied individuals.

Response 4: We have added a comparison in the discussion section to address this point. The following information was included: "Such low indicators of efficiency of technical preparedness of most players with musculoskeletal disorders compared to non-disabled individuals can be explained by low maneuverability in the choice of possible types of strokes." This revision can be found on page 10 of the revised manuscript.

Comment 5: Lines 357-359: Please indicate whether these changes were statistically significant and indicate the level of significance (i.e., p-values). Lines 392-395: Same comment.

Response 5: We have made the necessary changes to indicate statistical significance, including p-values and additional relevant statistical information. The changes are reflected in the text on pages 11 and 12.

Additional Text: "The significant finding of difference from constant in identifying two distinct types of billiard stroke techniques of 23 wheelchair users (one sample case): Type of power analysis - Sensitivity; Effect size (Cohen's d) - 0.5; α err prob - 0.05; Power (1-β err prob) - 0.80. The significant finding of difference from constant in identifying the technical-tactical and special physical preparedness level of 16 wheelchair users before and after modification equipment (one sample case): Type of power analysis - Sensitivity; Effect size (Cohen's d) - 0.5; α err prob - 0.05; Power (1-β err prob) - 0.65."

Comment 6: Please provide more detail on the characteristics of the athletes assessed throughout the studies described in the manuscript (i.e., males/females, ages, injury levels).

Response 6: Thank you for highlighting this point. We have expanded the details on the characteristics of the athletes assessed in our study, including gender, age, and injury levels. This information has been added to page 2 of the revised manuscript.

Updated Text: “Our pedagogical observation, a unique and crucial part of our research, analyzed the technical actions of 23 top-ranked European pool players with musculoskeletal disorders (38.5±1.87 years men). We focused on the model characteristics of billiard stroke techniques of wheelchair users. This observation led to a significant finding in identifying two distinct types of billiard stroke techniques for wheelchair users… Our pedagogical testing, a practical application of our research, assessed the technical-tactical and special physical preparedness of 16 Ukrainian billiard players with musculoskeletal disorders (12 male players: 37±5.5 years and four female players: 29.5±2.75 years).”

We have carefully considered all the changes the reviewer suggested and believe these revisions have significantly improved our manuscript. We appreciate the reviewer’s thorough review and constructive feedback.

Round 2

Reviewer 2 Report

Comments and Suggestions for Authors

The authors have responded and modified the aspects I highlighted in my initial review.

Although I had marked it as rejected at the time, the current modifications and, above all, the article's structuring and the discussion's improvement have improved the article sufficiently to recommend its publication.